# Evaluation of Transthoracic Echocardiography in the Assessment of Atherosclerosis of the Left Main Coronary Artery: Comparison with Optical Frequency Domain Imaging (a Pilot Study)

**DOI:** 10.3390/jcm10020256

**Published:** 2021-01-12

**Authors:** Fabien Labombarda, Vincent Roule, Idir Rebouh, Massimiliano Ruscica, Gerald F. Watts, Cesare R. Sirtori

**Affiliations:** 1Department of Cardiology, Centre Hospitalier Universitaire de Caen, 14000 Caen, France; fabienlabombarda@gmail.com (F.L.); roule-v@chu-caen.fr (V.R.); rebouh-i@chu-caen.fr (I.R.); 2Department of Pharmacological and Biomolecular Sciences, Università degli Studi di Milano, 20133 Milan, Italy; cesare.sirtori@icloud.com; 3Faculty of Health and Medical Sciences, School of Medicine, University of Western Australia, Perth, WA 6009, Australia; gerald.watts@uwa.edu.au; 4Lipid Disorders Clinic, Cardiometabolic Services, Department of Cardiology, Royal Perth Hospital, Perth, WA 6000, Australia; 5Centro Dislipidemie, ASST Grande Ospedale Metropolitano Niguarda Ca’ Granda, 20162 Milano, Italy

**Keywords:** atherosclerosis, echocardiography, left main coronary, optical frequency domain imaging

## Abstract

Background: Risk stratification using non-invasive imaging of the coronary vessels is emerging as an optimal standard of care for patients with dyslipidemias. Of particular interest is the evaluation of the left main coronary artery (LMCA), where calcium deposition appears to be a predictor of cardiovascular events. Methods: In coronary patients, we evaluated wall thickness and internal diameter of the LMCA examined by transthoracic echocardiography (TTE) and compared these with findings obtained by optical frequency domain imaging (OFDI), this latter also used to evaluate calcium deposition. Results: A significant positive correlation between TTE and OFDI for the anterior wall thickness (r = 0.41, *p* = 0.043) and internal diameter (r = 0.36, *p* = 0.048) of the LMCA was detected. Echocardiographic wall measurements were higher in patients with fibro-calcific plaques. The receiver operating characteristic (ROC) curve showed that an anterior wall thickness of LMCA ≥ 1.4 mm was predictive of fibro-calcific plaque (area under the curve = 0.815 and *p* = 0.006), sensitivity and specificity being 76.9% and 80%, respectively (Youden’s Index = 0.56). Conclusions: Measurement of anterior wall thickness of the LMCA by TTE and OFDI appears to be closely correlated and may predict the presence of coronary calcification.

## 1. Introduction

Severity of coronary artery disease is still best assessed by coronary angiography, although, over the past decade, many limitations have raised interest in intracoronary imaging modalities, such as intravascular ultrasound (IVUS), optical coherence tomography (OCT) and optical frequency domain imaging (OFDI) [1]. As an alternative to invasive imaging modalities, however, evaluation of coronary wall thickness may provide important information on the cardiovascular risk stratification [2]. Conventional non-invasive imaging methods, such as magnetic resonance imaging (MRI) or computed tomographic (CT) angiography, do not provide optimal resolution of wall thickness [3] and may require complex and expensive technologies, CT potentially exposing patients to radiation risk [4]. Measuring coronary wall thickness by transthoracic echography (TTE) may provide a useful alternative [5]. New high-quality probes for TTE allow, in fact, the evaluation of wall thickness of the left main coronary artery (LMCA), which appears to be significantly correlated to carotid intima-media thickness (C-IMT) [6], a well-established non-invasive tool to assess cardiovascular (CV) risk [7]. These findings supported the hypothesis that wall thickness of LMCA, determined by TTE, could predict coronary atherosclerosis and clinical events.

Recent reports have indicated that atherosclerosis of the LMCA is frequent in patients with familial hypercholesterolemia [8], as assessed by CT angiography [9] and that increased calcification of the LMCA appears to be a significant predictor of CV-specific and total mortality [10]. The present investigation explored the potential role of measurement of the wall thickness of the LMCA as a new non-invasive marker of coronary risk in patients, many of whom with hyperlipidemia. The overall objectives of the study, based on coronary patients undergoing angiography, were (1) to provide comparative data between TTE and optical frequency domain imaging (OFDI) for the assessment of the wall thicknesses of LMCA; (2) to associate wall thickness of the LMCA with coronary calcium as determined by OFDI.

## 2. Experimental Section

### 2.1. Patients

Between May 2016 and August 2018, consecutive coronary patients who underwent cardiac catheterization and OFDI including the LMCA were prospectively recruited for a specific echocardiographic analysis. The echocardiographic evaluation was performed within one month from OFDI. The protocol was approved by the Institutional Review Board of the University of Caen (2017-A01442-51).

### 2.2. Optical Frequency Domain Imaging Studies

The OFDI procedure (Lunawave^®^, FastView^®^, Terumo Europe N.V., Leuven, Belgium), performed as previously described [11], was carried out after intracoronary administration of nitroglycerin (0.2 mg). Images were acquired by using a non-occlusive technique (rate of 160 frames/s) during an automated pullback of the catheter (speed of 20 mm/s). The pullback was performed during continuous intracoronary injection of contrast medium through the guiding catheter using an injection pump (flow rate of 4 mL/s, maximum of 3 s). All images were recorded digitally, stored, and each frame read offline by two investigators (V.R. and I.R.) using previously validated criteria for OCT plaque characterization [12]. We analysed the whole LMCA from the ostium or catheter tip to the ostia of bifurcation branches defined as the first 5 mm of the left anterior descending artery and/or left circumflex artery. LM length was obtained from OFDI longitudinal reconstructions and defined as the distance between the first distal frame of the LM at the cross sectional image and the last proximal LM frame before aorta or catheter visualisation. When an atherosclerotic plaque was identified, at least three measurements of the intima and media thickness were performed where the plaque was largest. Quantitative coronary angiography was performed offline (CAAS II, Pie Medical, Maastricht, The Netherlands) using validated quantitative methods.

### 2.3. Echocardiographic Studies

Examinations were obtained using a commercially available ultrasound system (EPIC 7; Philips Medical Systems, Bothell, WA, USA) with a broadband phased array sector high frequency transducer (S8-3) with 3–8 MHz frequency range resulting in an axial resolution 0.25 mm. The TTE evaluation was conducted by one experienced physician (FL) blinded to the OFDI results. Each subject was examined in a semi-supine left lateral position. The electrocardiogram was recorded continuously. The LMCA was visualized using the standard parasternal short-axis view, from the second or third intercostal space, by scanning superior to the aortic valve with careful interrogation of the aortic sinuses. LMCA generally arises approximately at 4 o’clock considering the aortic root as a clock face [6]. After the origin of the LMCA was identified, the image was optimized by step-by-step movement of the transducer according to the course of the vessel in order to visualize the LMCA in its entire extension including the bifurcations (Figure 1B). Machine controls were optimized by the operator at each examination. Optimization included altering the depth, movement of focal zone to region of the LMCA, reduction of sector size, alteration of gain control and dynamic range. Color Doppler mapping was systemically used to confirm visualization of the LMCA (velocity range was set with a low Nyquist limit 15–20 cm/s, filters were reduced, and priority was increased). For all patients, LMCA measurements were made in the same area, i.e., in the first 10–15 mm after the origin from the aorta, on the frame where the LMCA was best visualized (Figure 1A). Measurements of the internal diameter, anterior and posterior wall thicknesses of the LMCA were performed. The left ventricular ejection fraction was assessed from an apical 4 and 2 chambers view using the modified biplane Simpson’s rule following the joint guidelines of the European Association of Echocardiography and the American Society of Echocardiography [13].

### 2.4. Statistical Analysis

Continuous and categorical variables were expressed as mean ± standard deviation and numbers of patients and percentages. Comparison between the two groups was performed using a Mann–Whitney test, and correlations by using the Pearson coefficients. The receiver operating characteristic (ROC) curves were analysed to assess the best cut-off of wall thicknesses of the LMCA to predict fibro-calcific plaque. Statistical significance was defined as *p* < 0.05.

## 3. Results

A total of 32 consecutive patients were enrolled. Adequate TTE analysis of the LMCA was not possible in seven subjects because of sub-optimal echogenicity. As reported in Table 1, 76% of the remaining 25 patients (age 58 ± 15 years) were males. Body mass index (BMI) was 25.6 ± 4.2 Kg/m^2^ with two individuals in the range of obesity (BMI > 30 Kg/m). Total- and LDL-C were at moderately elevated mean levels, 76% of the patients being on statin treatment. Total and LDL-C were comparable between groups: 4.82 ± 1.54 and 3.1 ± 1.42 mmol/L, respectively, in patients with fibro calcific plaques and 4.38 ± 1.31 and 2.58 ± 1.34 mmol/L, respectively, in patients without fibro-calcific plaques. Most of the patients presented with ST-elevation myocardial infarction (*n* = 20, 80%) and were characterized by a single vessel disease (*n* = 17, 68%); 32% had angiographically multivessel diseases. Angiographic signs of atherosclerosis of the LMCA, i.e., lesions ≥ 30% were detected in 40% of patients.

OFDI showed a mean thickness of the LMCA of 0.38 ± 0.21 mm, mean diameter 4.3 ± 0.4 mm and mean lumen area 13.1 ± 3.9 mm^2^. Fibro-calcific plaques were detected in 76% of the patients (*n* = 19). The evaluation of LMCA by TTE led to the following results: mean anterior wall thickness 1.5 ± 0.2 mm, mean posterior wall thickness 1.4 ± 0.3 mm, mean internal diameter 4 ± 0.8 mm and mean external diameter 6.2 ± 2 mm. Intima-media thickness evaluated by OFDI and anterior wall thickness of the LMCA evaluated by TTE showed a positive correlation (r = 0.41, *p* = 0.043) similar to the internal diameter (r = 0.36, *p* = 0.048). Internal diameter by TTE was significantly correlated with the anterior wall thickness by TTE (r: 0.61, *p* < 0.01) (Figure 2).

Echocardiographic measurements of the internal diameter and anterior wall thickness of the LMCA were significantly higher in patients with fibro-calcific plaques (Figure 3A,B). The diagnostic performance of wall thickness of the LMCA for the detection of fibro-calcific plaques is presented as an ROC curve (Figure 3C). A value > 1.4 mm was associated with fibro-calcific plaque (area under the curve: 0.815; *p* = 0.006), with the highest sensitivity and specificity of 76.9% and 80%, respectively, to predict fibro-calcific plaques of the LMCA. Youden’s Index was 0.56.

## 4. Discussion

A positive association between the anterior wall thickness of the LMCA evaluated by invasive and non-invasive approaches, OFDI and TTE, was observed, also allowing to detect a threshold thickness of the LMCA, associated with increased fibro-calcific plaques. The association between calcification of the LMCA and CV risk was earlier attributed to the occurrence of frequent lesions in this coronary segment [14] with a CV risk in the same range as predicted by the global calcium score. More recently, in a study evaluating features and distribution of plaques in patients with and without genetically confirmed familial hypercholesterolemia, Pang et al. [9] reported an increased frequency of coronary calcium in the LMCA, left anterior descending artery and right coronary artery, indicating the need to assess CT cardiac scanning in these affected sites. The presented observations allow predicting the presence of calcifications at one of these major sites and to assess the clinical severity of disease, also simplifying the clinical approach with minimal cost. Furthermore, the risk prediction based on the calcium score, area and density, does not appear to be modified by statin treatment [15].

The non-invasive imaging procedure of the coronary arteries [3] by TTE for coronary diagnostics may be also successfully used to detect coronary artery abnormalities [16,17] and coronary reserve. Use of TTE for CV risk assessment has been supported by the correlation between wall thickness of LMCA and C-IMT, a reliable tool for risk determination [7]. As recently pointed out by Meah et al. [18], C-IMT may not always provide a reliable assessment of risk prediction, although C-IMT changes over time appear to be associated with CV risk changes in the same direction [19].

The strengths of the present study were the sensitivity of this coronary segment to the damage exerted by lipid disorders [8] and, more so, by calcium deposition, considering the potential of LMCA calcifications to predict coronary events [10]. The reported investigation evaluated wall thickness of the LMCA in parallel by TTE and OFDI, a sensitive tool for the direct assessment of the coronary structural information and fibro-calcific plaque characterization [2]. There is as yet non final evidence of a superior performance of OCT vs. IVUS as a guide for coronary interventions [20], this being the objective of the ongoing OCTIVUS Trial [21].

In addition, to confirm the reliability of coronary wall thickness determined by TTE as a marker of disease is the observation that, above a value of 1.4 mm of the anterior wall thickness of the LMCA, there is an increased prevalence of calcified plaques, identified as a significant predictor of CV specific and total mortality [10]. Interestingly, with a magnetic resonance imaging methodology, Ghanem et al. [2] reported that, above a mean threshold of 1.41 mm for coronary wall thickness (evaluated in the right coronary), there is a substantial increase of coronary artery disease risk in women (coronary segment score > 5); in addition, they reported a high correlation between coronary calcium score and coronary WT. The TTE methodology appears thus to provide findings similar to a high-cost technology, being also indicative of an increased CV risk. More recently, CV risk assessment from coronary artery calcium has been shown to improve decisions on drug treatment for familial hypercholesterolemia [22]. The absence of coronary calcium can provide a potentially useful risk marker: a coronary artery calcium (CAC) = 0 in a patient with familial hypercholesterolemia appears to be associated with essentially no rise in coronary risk [22,23], providing a clinically more significant information compared to polygenic risk scores [24]. Finally, coronary calcium has been associated with both atherosclerotic cardiovascular diseases and bleeding, identifying individuals experiencing net benefit from primary prevention with aspirin therapy [25]. As indicated by Pugliese et al., spotty calcification patterns and attenuation, not associated with stenosis, are both linked to a higher degree of plaque instability [26]. These are identifiable with both OFDI and TTE. The latter approach, even at a single major coronary segment, can be thus of help in an initial diagnostic approach to the patient.

Limitations of the present study, which has to be viewed as a pilot or proof-of-principle study, are the small size of the sampled population, that comprised Caucasians from Continental Europe. Lumen area may be somewhat overestimated by the OFDI technology because the use of nitroglycerin during the procedure may lead to coronary dilatation, also likely due to vascular remodeling. A further weakness of this approach could be the potential poor image quality in some patients, essentially dependent on a thickened anterior chest wall in obese individuals or breasts in women, and trembling consequent to drug treatment. These can be overcome by training and more advanced equipment. Finally, reproducibility of the wall thickness of LMCA by TTE in another group of coronary individuals should be tested.

## 5. Conclusions

In conclusion, the wider use of TTE for the evaluation of wall thickness of the LMCA may provide a sensitive approach to improve risk prediction in patients who require further evaluation. This small pilot study, limited by the access of patients with available OFDI to undergo TTE evaluation, (1) has shown agreement between TTE and OFDI in the assessment of wall thickness of the LMCA and (2) tentatively suggests that the measurement of wall thickness may predict the presence of coronary calcification; more extensive studies are required to verify this notion. The use of echography to stratify risk can also provide information on the ventricular structure, thickness and wall motion abnormalities, including aortic valve stenosis, severity and progression. In our institutions, vascular monitoring by C-IMT and by TTE with a special interest for hyperlipidemic patients has generally confirmed the associated CV risk.

## Figures and Tables

**Figure 1 jcm-10-00256-f001:**
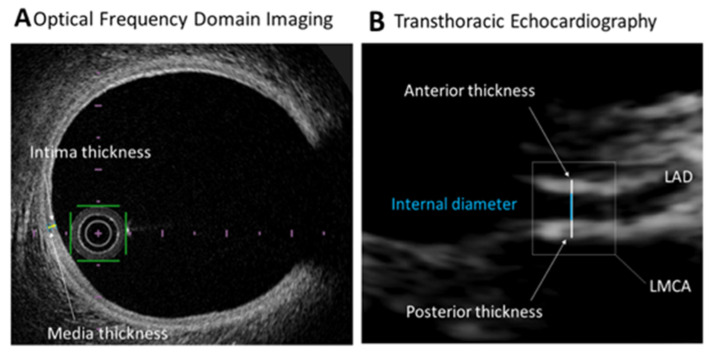
(**A**,**B**) LMCA measurements by optical frequency domain imaging (OFDI) and transthoracic echocardiography (TTE). LAD: left anterior descending, LMCA: left main coronary artery.

**Figure 2 jcm-10-00256-f002:**
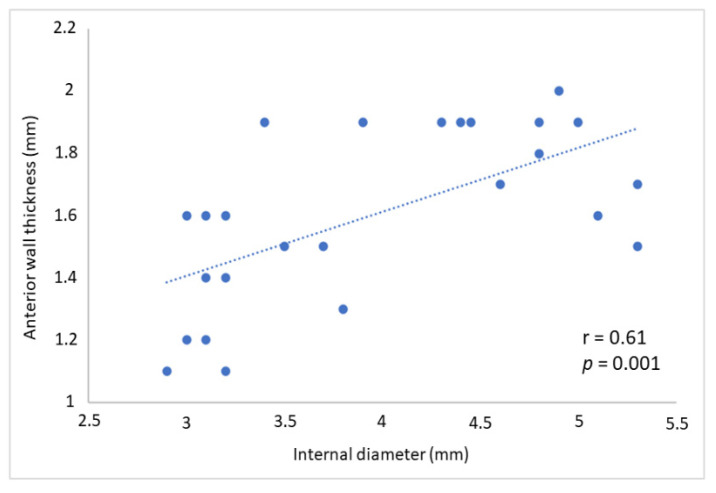
Correlation between anterior wall thickness and internal diameter of the LMCA measured by TTE. LMCA: left main coronary artery; TTE: trans thoracic echocardiogram.

**Figure 3 jcm-10-00256-f003:**
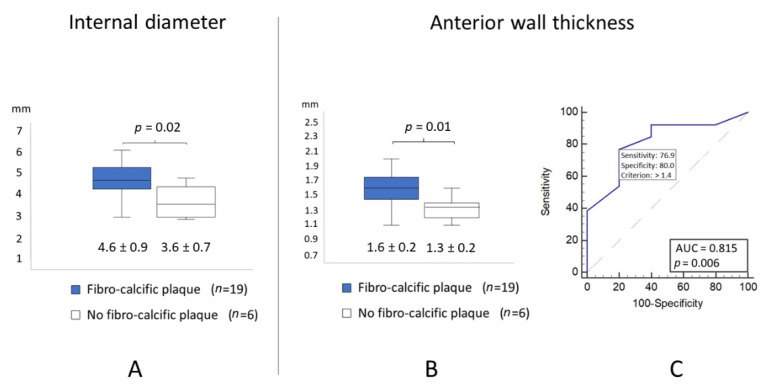
(**A**) internal diameter of the LMCA measured by TTE in patients with and without fibro calcific plaques (4.6 ± 0.9 mm vs. 3.6 ± 0.7 mm, respectively); (**B**) anterior wall thickness of the LMCA measured by TTE in patients with and without fibro calcific plaques (1.6 ± 0.2 mm vs. 1.3 ± 0.2 mm, respectively); (**C**) ROC curves for predicting the presence of fibro-calcific plaques by TTE measurement of the anterior wall thickness of LMCA. LMCA: left main coronary artery; OFDI: optical frequency domain imaging; ROC: receiver operating characteristic; TTE: transthoracic echography.

**Table 1 jcm-10-00256-t001:** Demographic and angiographic characteristics of the population study.

Variables	Population (*n* = 25)
**Baseline characteristics**	
Age (years)	58 ± 15
Gender (Men %)	19 (76%)
Body mass index (kg/m^2^)	25.6 ± 4.2
Body mass index > 30 kg/m^2^	2 (8%)
Systemic hypertension	7 (28%)
Hyperlipidemia	10 (40%)
Current smokers	13 (52%)
Diabetes mellitus	1 (4%)
Hyperlipidemia	10 (39%)
Total cholesterol (mmol/L)	4.9 ± 1.8
Low-density lipoprotein (mmol/L)	3.2 ± 1.7
High-density lipoprotein (mmol/L)	1.4 ± 1.1
Triglycerides (mmol/L)	2.0 ± 1.4
Patients on statin treatment	19 (76%)
Myocardial Infarction	0 (0%)
Coronary artery bypass graft	0 (0%)
Percutaneous coronary intervention	1 (4%)
Estimated glomerular filtration rate < 60 mL/min	1 (4%)
Left ventricular ejection fraction (%)	54 ± 9
**Clinical presentation**	
Stable angina	1 (4%)
Non ST-elevation myocardial infarction	4 (16%)
ST-elevation myocardial infarction	20 (80%)
**Angiographic characteristics**	
Guiding catheter	
Extra Back-Up	22 (88%)
Judkins Left guiding catheter	3 (12%)
Single-vessel disease	17 (68%)
Two-vessel disease	2 (8%)
Three-vessel disease	6 (24%)
LMCA length (mm)	12 ± 4
LMCA reference diameter (mm)	4 ± 0.6
**Angiographic signs of LMCA atherosclerosis**	
No lesions	15 (60%)
Lesions ≥ 30%	10 (40%)

LMCA = left main coronary artery.

## Data Availability

The datasets used and/or analysed during the current study are available from Fabien Labombarda upon reasonable request.

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
