# Peer review of "Evaluation of Transthoracic Echocardiography in the Assessment of Atherosclerosis of the Left Main Coronary Artery: Comparison with Optical Frequency Domain Imaging (a Pilot Study)"

_jcm, 2021, doi:10.3390/jcm10020256_

Round 1
Reviewer 1 Report
The authors compared TTE assessment of LMCA disease using OFDI as the gold standard. The problem is that OFDI is not the gold standard for LMCA assessment. OFDI (or OCT in general) is limited by the size of the LMCA (in some patients) and by the inability to image the ostial because of the problem with blood clearance and flushing. While the authors mention limitations of TTE and the number of patients in whom TTE was suboptimal, they do not provide similar information regarding OCT. The fact that only 25 patients were enrolled in more than 2 years highlights this issue. The conclusions represent a huge over-reach. At the most, this is a feasibility study.
The patient population is highly skewed toward STEMI/NSTEMI patients. If the purpose of TTE assessment of the LMCA is to identify patients with CAD who would benefit from risk factor control, they should include such patients. STEMI/NSTEMI patients obviously have CAD.
How did the authors image the ostium of the LMCA with OCT? The authors stated: "The whole LMCA (from the ostium or catheter tip to the ostia of its bifurcation branches defined as the first 5 mm of the left anterior descending artery and/or left circumflex artery) was analysed." I am skeptical that they were able to image the ostium of the LMCA in 32 patients. Also, the authors never provided the methodology used to measure LMCA wall thickness measurements; and they should provide a more complete output of the OCT analysis of the LMCA and proximal LAD.
How complete was the TTE assessment of the LMCA? Did it always include the bifurcation? It is noteworthy that there is a lot of detail regarding TTE, but a paucity of data regarding OCT methodology.
There are no angiographic methods.
The authors second purpose is "to associate wall thickness of LMCA with coronary calcium as determined by OFDI." However, they never report OCT coronary calcium. They do report the association between TTE wall thickness and fibrocalcific plaque, but not coronary calcium. In addition, they never report OCT quantification of either coronary calcium or fibrocalcific plaque. Finally, there were too few patients without fibrocalcific plaque (n=6) for this type of statistical comparison. The authors stated: "The diagnostic 1performance of wall thickness of the LMCA for the detection of fibro-calcific plaque is presented as a ROC curve (Figures 1C). A value >1.4 mm was associated with fibro-calcific plaque (area under the curve: 0.815; p=0.006), with the highest sensitivity and specificity of 76.9% and 80%, respectively, to predict fibro-calcific plaques of the LMCA. Youden’s Index was 0.56." Where was the fibrocalcific plaque? In the LMCA or elsewhere? Why is this important? If there is fibrocalcific plaque in the LMCA or increased LMCA all thickness, doesn't the patient have CAD?
Author Response
The authors compared TTE assessment of LMCA disease using OFDI as the gold standard. The problem is that OFDI is not the gold standard for LMCA assessment. OFDI (or OCT in general) is limited by the size of the LMCA (in some patients) and by the inability to image the ostial because of the problem with blood clearance and flushing. While the authors mention limitations of TTE and the number of patients in whom TTE was suboptimal, they do not provide similar information regarding OCT. The fact that only 25 patients were enrolled in more than 2 years highlights this issue. The conclusions represent a huge over-reach. At the most, this is a feasibility study.
Thank you for the comment. The Reviewer will appreciate that OCT evaluations certainly have limitations. However, as indicated in Ref.1 (Kang et al., 2020), there is no general consensus and, in the Introduction of the ongoing OCTIVUS study, that compares OCT and IVUS, it is rightfully pointed out that OCT has a “superior spatial resolution” and that study attempts to give a clinical meaning to this important concept. In our study, we tried to obtain a meaningful comparative evaluation between the TTE and OCT, this choice being dictated by the superior resolution of the latter. The technical criticism raised, was not a major consideration in the present study, where we simply wished to provide further information on this technology, to be eventually used in clinical intervention studies requiring multiple evaluations.
Concerning the difficulties in imaging LMCA with OCT, we agree with the Reviewer’s comment. OCT was not considered suitable for the assessment of the LMCA, because of the large coronary size and poor blood washing. The recently developed OFDI provides higher acquisition speed and larger field of view compared with prior generation time-domain OCT, which may potentially overcome previous limitations. Our team recently demonstrated that OFDI can accurately evaluate the LM and detect and assess angiographically non visualized atherosclerotic plaques, providing accurate assessment of >90% of the quadrants of the LM and the ostia of its bifurcation branches (Roule V et al, 2020).
Overall, our work has implications for clinical intervention studies (e.g., with lipid lowering medications), where a non-invasive technique is certainly more suitable for repeated assessments. We emphasize that we selected patients with well visible LMCA at OFDI and we examined them by TTE.
We have now revised the over-reaching nature of our previous conclusion: In conclusion, the wider use of the TTE for the evaluation of the wall thickness of LMCA may provide a sensitive strategy to improve risk prediction, e.g., in the case of hyperlipidemic patients. This small pilot study, limited by the access of patients with available OFDI to undergo TTE evaluation, (1) has shown agreement between TTE and OFDI in the assessment of wall thickness of the LMCA and (2) tentatively suggests that the measurement of wall thickness may predict the presence of coronary calcification; more extensive studies are required to verify this notion. The use of echography to stratify risk can also provide information on the ventricular structure, thickness and wall motion abnormalities, including aortic valve stenosis, severity and progression. In our institutions vascular monitoring by C-IMT and by TTE with a special interest for hyperlipidemic patients, has generally confirmed the associated CV risk.
References
Kang, D.Y.; Ahn, J.M.; Park, H.; Lee, P.H.; Kang, S.J.; Lee, S.W.; Kim, Y.H.; Park, S.W.; Kim, S.W.; Hur, S.H., et al. Comparison of optical coherence tomography-guided versus intravascular ultrasound-guided percutaneous coronary intervention: Rationale and design of a randomized, controlled OCTIVUS trial. Am Heart J 2020, 228, 72-80, doi:10.1016/j.ahj.2020.08.003.
Roule V, Rebouh I, Lemaitre A, Bignon M, Ardouin P, Sabatier R, Labombarda F, Blanchart K, Beygui F. Evaluation of Left Main Coronary Artery Using Optical Frequency Domain Imaging and Its Pitfalls. J Interv Cardiol. 2020 Jun 12;2020:4817239. doi: 10.1155/2020/4817239. PMID: 32581660; PMCID: PMC7306070.
The patient population is highly skewed toward STEMI/NSTEMI patients. If the purpose of TTE assessment of the LMCA is to identify patients with CAD who would benefit from risk factor control, they should include such patients. STEMI/NSTEMI patients obviously have CAD.
As above indicated, the study followed the opposite route, i.e., that of recruiting patients with well visible OFDI and with ischemic disease (80% STEMI, 16% NSTEMI) as reported in Table 1. This is what we found. There is no question such patients need risk factor control.
How did the authors image the ostium of the LMCA with OCT? The authors stated: "The whole LMCA (from the ostium or catheter tip to the ostia of its bifurcation branches defined as the first 5 mm of the left anterior descending artery and/or left circumflex artery) was analysed." I am skeptical that they were able to image the ostium of the LMCA in 32 patients. Also, the authors never provided the methodology used to measure LMCA wall thickness measurements; and they should provide a more complete output of the OCT analysis of the LMCA and proximal LAD.
The OFDI procedures were performed according the methods described in the Expert Review Document published by Prati F et al.
We have now added some details regarding the OFDI method as follows: ‘All images were recorded digitally, stored, and each frame read off-line by 2 investigators (VR and IR) using previously validated criteria for OFDI plaque characterization (Jia et al., 2013). We analyzed the whole LMCA from the LMCA ostium or catheter tip to the ostia of its bifurcation branches, defined as the first 5mm of the left anterior descending artery and/or left circumflex artery. LM length was obtained from the OFDI longitudinal reconstructions and defined as the distance between the first distal frame of the LM at cross sectional image and the last proximal LM frame before aorta or catheter visualization. When atherosclerotic plaque was identified, at least 3 measurements of the intima and media thickness were performed where the plaque was largest.’
References
Prati F, Regar E, Mintz GS, Arbustini E, Di Mario C, Jang IK, Akasaka T, Costa M, Guagliumi G, Grube E, Ozaki Y, Pinto F, Serruys PW; Expert's OCT Review Document. Expert review document on methodology, terminology, and clinical applications of optical coherence tomography: physical principles, methodology of image acquisition, and clinical application for assessment of coronary arteries and atherosclerosis. Eur Heart J. 2010 Feb;31(4):401-15. doi: 10.1093/eurheartj/ehp433. Epub 2009 Nov 4. PMID: 19892716.
Jia H, Abtahian F, Aguirre AD, Lee S, Chia S, Lowe H, Kato K, Yonetsu T, Vergallo R, Hu S, Tian J, Lee H, Park SJ, Jang YS, Raffel OC, Mizuno K, Uemura S, Itoh T, Kakuta T, Choi SY, Dauerman HL, Prasad A, Toma C, McNulty I, Zhang S, Yu B, Fuster V, Narula J, Virmani R, Jang IK. In vivo diagnosis of plaque erosion and calcified nodule in patients with acute coronary syndrome by intravascular optical coherence tomography. J Am Coll Cardiol. 2013 Nov 5;62(19):1748-58. doi: 10.1016/j.jacc.2013.05.071. Epub 2013 Jun 27. PMID: 23810884; PMCID: PMC3874870.
How complete was the TTE assessment of the LMCA? Did it always include the bifurcation? It is noteworthy that there is a lot of detail regarding TTE, but a paucity of data regarding OCT methodology. There are no angiographic methods.
TTE assessment: The bifurcation was always included. We revised the sentence as follows: ‘After the origin of the LMCA was identified, the image was optimized by step-by-step movement of the transducer according to the course of the vessel in order to visualize the LMCA in its entire extension including the bifurcation’ (Figure 1B).
Angiographic methods: we revised the sentence in the Methods section as follows: “Quantitative Coronary Angiography. Quantitative coronary angiography was performed offline (CAAS II,Pie Medical, Maastricht, Netherlands) using validated quantitative methods (Garrone et al., 2009).
Reference
Garrone P, Biondi-Zoccai G, Salvetti I, Sina N, Sheiban I, Stella PR, Agostoni P. Quantitative coronary angiography in the current era: principles and applications. J Interv Cardiol. 2009 Dec;22(6):527-36. doi: 10.1111/j.1540-8183.2009.00491.x. Epub 2009 Jul 13. PMID: 19627430.
The authors second purpose is "to associate wall thickness of LMCA with coronary calcium as determined by OFDI." However, they never report OCT coronary calcium. They do report the association between TTE wall thickness and fibrocalcific plaque, but not coronary calcium. In addition, they never report OCT quantification of either coronary calcium or fibrocalcific plaque. Finally, there were too few patients without fibrocalcific plaque (n=6) for this type of statistical comparison. The authors stated: "The diagnostic 1performance of wall thickness of the LMCA for the detection of fibro-calcific plaque is presented as a ROC curve (Figures 1C). A value >1.4 mm was associated with fibro-calcific plaque (area under the curve: 0.815; p=0.006), with the highest sensitivity and specificity of 76.9% and 80%, respectively, to predict fibro-calcific plaques of the LMCA. Youden’s Index was 0.56." Where was the fibrocalcific plaque? In the LMCA or elsewhere? Why is this important? If there is fibrocalcific plaque in the LMCA or increased LMCA all thickness, doesn't the patient have CAD?
Studied patients had definite CAD Since one of us (GW) had reported that in hypercholesterolemic pts with CVD there was calcium deposition and this might be linked to raised CV risk (Lahti et al., 2019), we tested the hypothesis, that turned out to be probably right, that coronary calcification is associated with raised LMCA thickness by TTE. The predictive role of raised calcium deposition is supported by a larger study evaluating long term CV risk (Garcia-Garcia et al., 2014) having to do specifically with the deposition of calcium in the LMCA, not elsewhere.
The limits of this type of statistical evaluation are well known. Actually, in the paper in Ref. 2 (Ghanem et al., 2019), where coronary thickness was determined by MRI, a threshold of 1.41 mm is reported for the presence of CAD. In this paper the Authors repeatedly underline the correlation of CAD scores with calcium deposition.
In addition, interest is focusing on non-stenotic, spotty calcifications and attenuation that correlate with markers of plaque instability. Most recently the issue of coronary calcification has grown in interest after the observation that a calcium score of 0 in familial hypercholesterolemic patients was associated with no rise in CV risk. (Mszar 2020 (a) and Mszar 2020 (b)). The text has been revised according and now reads as follows “The absence of coronary calcium can provide a potentially useful risk marker: a CAC= 0 in a patient with familial hypercholesterolemia appears to be associated with essentially no rise in coronary risk [22,23], providing a clinically more significant information compared to polygenic risk scores [24].”
Reference
Lahti, S.J.; Feldman, D.I.; Dardari, Z.; Mirbolouk, M.; Orimoloye, O.A.; Osei, A.D.; Graham, G.; Rumberger, J.; Shaw, L.; Budoff, M.J., et al. The association between left main coronary artery calcium and cardiovascular-specific and total mortality: The Coronary Artery Calcium Consortium. Atherosclerosis 2019, 286, 172-178, doi:10.1016/j.atherosclerosis.2019.03.015.
Garcia-Garcia, H.M.; Muramatsu, T.; Nakatani, S.; Lee, I.S.; Holm, N.R.; Thuesen, L.; van Geuns, R.J.; van der Ent, M.; Borovicanin, V.; Paunovic, D., et al. Serial optical frequency domain imaging in STEMI patients: the follow-up report of TROFI study. Eur Heart J Cardiovasc Imaging 2014, 15, 987-995, doi:10.1093/ehjci/jeu042.
Ghanem, A.M.; Matta, J.R.; Elgarf, R.; Hamimi, A.; Muniyappa, R.; Ishaq, H.; Hadigan, C.; McConnell, M.V.; Gharib, A.M.; Abd-Elmoniem, K.Z. Sexual Dimorphism of Coronary Artery Disease in a Low- and Intermediate-Risk Asymptomatic Population: Association with Coronary Vessel Wall Thickness at MRI in Women. Radiol Cardiothorac Imaging 2019, 1, e180007, doi:10.1148/ryct.2019180007.
[22] Mszar, R.; Nasir, K.; Santos, R.D. Coronary Artery Calcification in Familial Hypercholesterolemia: An Opportunity for Risk Assessment and Shared Decision Making With the Power of Zero? Circulation 2020, 142, 1405-1407, doi:10.1161/CIRCULATIONAHA.120.049057.
[23] Mszar, R.; Grandhi, G.R.; Valero-Elizondo, J.; Virani, S.S.; Blankstein, R.; Blaha, M.; Mata, P.; Miname, M.H.; Al Rasadi, K.; Krumholz, H.M., et al. Absence of Coronary Artery Calcification in Middle-Aged Familial Hypercholesterolemia Patients Without Atherosclerotic Cardiovascular Disease. JACC Cardiovasc Imaging 2020, 13, 1090-1092, doi:10.1016/j.jcmg.2019.11.001.
[24] Wunnemann, F.; Sin Lo, K.; Langford-Avelar, A.; Busseuil, D.; Dube, M.P.; Tardif, J.C.; Lettre, G. Validation of Genome-Wide Polygenic Risk Scores for Coronary Artery Disease in French Canadians. Circ Genom Precis Med 2019, 12, e002481, doi:10.1161/CIRCGEN.119.002481.
Reviewer 2 Report
Although the idea of ​​this study is very interesting, some aspects should improve: 1) The conclusions should respond to the two initial hypotheses.
2) In line 92 in the Simpson method the 2-chamber view of the left ventricle should be commented.
3) Although significant, Pearson's correlation coefficients seem very low.
4) As the authors indicate, the sample size is not sufficient to reach solid conclusions.
Author Response
Although the idea of this study is very interesting, some aspects should improve:
1) The conclusions should respond to the two initial hypotheses.
Yes, we changed the conclusions accordingly. As requested by the other Referee we have revised the text of our conclusion. The sentence now reads as follows “In conclusion, the wider use of the TTE for the evaluation of the wall thickness of LMCA may provide a sensitive strategy to improve risk prediction, e.g., in the case of hyperlipidemic patients. This small pilot study, limited by the access of patients with available OFDI to undergo TTE evaluation, (1) has shown agreement between TTE and OFDI in the assessment of wall thickness of the LMCA and (2) tentatively suggests that the measurement of wall thickness may predict the presence of coronary calcification; more extensive studies are required to verify this notion. The use of echography to stratify risk can also provide information on the ventricular structure, thickness and wall motion abnormalities, including aortic valve stenosis, severity and progression. In our institutions vascular monitoring by C-IMT and by TTE with a special interest for hyperlipidemic patients, has generally confirmed the associated CV risk.”
2) In line 92 in the Simpson method the 2-chamber view of the left ventricle should be commented.
We agree with this comment. We revised the sentence as follows: ‘The left ventricular ejection fraction was assessed from an apical 4 and 2chambers view using the modified biplane Simpson’s rule, according to the joint guidelines of the European Association of Echocardiography and the American Society of Echocardiography (Lang et al., 2015),
Reference
Lang RM, Badano LP, Mor-Avi V, Afilalo J, Armstrong A, Ernande L, Flachskampf FA, Foster E, Goldstein SA, Kuznetsova T, Lancellotti P, Muraru D, Picard MH, Rietzschel ER, Rudski L, Spencer KT, Tsang W, Voigt JU. Recommendations for cardiac chamber quantification by echocardiography in adults: an update from the American Society of Echocardiography and the European Association of Cardiovascular Imaging. J Am Soc Echocardiogr. 2015 Jan;28(1):1-39.e14. doi: 10.1016/j.echo.2014.10.003. PMID: 25559473.
3) Although significant, Pearson's correlation coefficients seem very low.
We agree but the numbers were low. These are patients who had a positive angiography with OFDI and then sent for TTE. As well understandable to any Cath lab specialist, this cuts down numbers considerably
4) As the authors indicate, the sample size is not sufficient to reach solid conclusions.
Even some statin trials, such as the IDEAL Study, with huge numbers of patients, did not reach a solid conclusion! We believe that these non-invasive techniques have a positive future and even small studies such as this, may be of value; our investigation has to be viewed as a pilot or proof-of-principle study. The limitation section now reads as follows “Limitations of the present study, which has to be viewed as a pilot or proof-of-principle study, are the small size of the sampled population, who comprised Caucasians from Continental Europe. Lumen area may be somewhat overestimated by the OFDI technology, because the use of nitroglycerin during the procedure may lead to dilatation of coronary diameter in atheromatous sections and also likely due to vascular remodeling A further weakness of this approach could be the potential poor image quality in some patients, essentially dependent on a thickened anterior chest wall in obese individuals or breasts in women, and trembling consequent to drug treatment. These can be overcome by training and more advanced equipment.”
Reviewer 3 Report
In their study entitled “Evaluation of transthoracic echocardiography in the assessment of atherosclerosis of left main coronary artery: comparison with optical frequency domain imaging” Labombarda et al. present a novel approach to assess cardiovascular risk by measuring the anterior wall thickness of the left main coronary artery (LMCA) by transthoracic echocardiography (TTE). A value > 1.4 mm correlated with the presence of atheroma detected by optical frequency domain imaging (ODFI).
The approach is novel and of clinical importance. The article is well written and comprehensive.
Major issues:
At what frequency operated the TTE probe? Standard probes are set to 3-5 MHz – resulting in an axial resolution of about 0.3-0.6 mm. This would imply that measuring one more or one less pixel would change the result substantially when measuring such small anatomical structures as the LMCA wall. Did the authors analyze the intra- and interobserver variability? It appears noteworthy that the authors compared the entire will thickness measured by TTE with the intima media thickness measured by OFDI. Do the authors have comparable data obtained by intravascular ultrasound (IVUS) which might allow comparing measurements of the thickness of the entire LCMA wall by two similar modalities?
It would also be helpful to see a scatter plot of the correlations of the wall thickness and the internal diameter of the LMCA.
Can the authors explain why the internal diameter is higher in patients with atheroma? Remodeling?
Minor issues:
On page 4, lines 123-125 the authors refer to Figure 1 when, in fact, they should refer to Figure 2.
The sentence “There is as yet non final evidence on a superior performance of OCT vs IVUS as a guide for coronary intervention [17] and this being the objective of the ongoing OCTIVUS Trial [1].” (page 5, lines 157-159) seems out of context. The best imaging modality for PCI guidance does not appear to be the scope of the article.
Author Response
Reviewer#3
In their study entitled “Evaluation of transthoracic echocardiography in the assessment of atherosclerosis of left main coronary artery: comparison with optical frequency domain imaging” Labombarda et al. present a novel approach to assess cardiovascular risk by measuring the anterior wall thickness of the left main coronary artery (LMCA) by transthoracic echocardiography (TTE). A value > 1.4 mm correlated with the presence of atheroma detected by optical frequency domain imaging (ODFI).
The approach is novel and of clinical importance. The article is well written and comprehensive.
Major issues:
At what frequency operated the TTE probe?
We used a broadband, phased array sector high frequency transducer (S8-3) with a 3-8 MHz frequency range resulting in an axial resolution of 0.25 mm. This information is now given in section 2.3 “Echocardiographic studies”.
Standard probes are set to 3-5 MHz – resulting in an axial resolution of about 0.3-0.6 mm. This would imply that measuring one more or one less pixel would change the result substantially when measuring such small anatomical structures as the LMCA wall. Did the authors analyze the intra- and interobserver variability? It appears noteworthy that the authors compared the entire will thickness measured by TTE with the intima media thickness measured by OFDI.
Data on intra- and inter-observer variabilities are reported in the following Table. Since this issue was covered in prior papers from our groups, we elected not to add this additional material to a paper that was already above allowed limits.
Variables |
Intra observer reproducibility Intra Class correlation (95% Confident Interval)
|
Inter observer reproducibility Intra Class Correlation (95% Confident Interval) |
Anterior thickness |
0.98 (0.88 - 0.98) |
0.93 (0.76 - 0.99) |
Internal diameter |
0.96 (0.85 - 0.99) |
0.91 (0.80- 0.96) |
Do the authors have comparable data obtained by intravascular ultrasound (IVUS) which might allow comparing measurements of the thickness of the entire LCMA wall by two similar modalities?
We have no data on IVUS in these patients
It would also be helpful to see a scatter plot of the correlations of the wall thickness and the internal diameter of the LMCA. FB can you do it?
Figure 3: Correlation between anterior wall thickness and internal diameter measured by trans thoracic echocardiogram
We revised the results section as follows: ‘Internal diameter by TTE was significantly correlated with the anterior wall thickness by TTE (r: 0.61, p < 0.01)’
Can the authors explain why the internal diameter is higher in patients with atheroma? Remodeling?
This is most likely due to vascular remodeling and have now stated in this on the discussion and now reads as follows “Limitations of the present study, which has to be viewed as a pilot or proof-of-principle study, are the small size of the sampled population, who comprised Caucasians from Continental Europe. Lumen area may be somewhat overestimated by the OFDI technology, because the use of nitroglycerin during the procedure may lead to dilatation of coronary diameter in atheromatous sections and also likely due to vascular remodeling. A further weakness of this approach could be the potential poor image quality in some patients, essentially dependent on a thickened anterior chest wall in obese individuals or breasts in women, and trembling consequent to drug treatment. These can be overcome by training and more advanced equipment.”
Minor issues:
On page 4, lines 123-125 the authors refer to Figure 1 when, in fact, they should refer to Figure 2.
Apologies, now corrected.
The sentence “There is as yet non final evidence on a superior performance of OCT vs IVUS as a guide for coronary intervention [17] and this being the objective of the ongoing OCTIVUS Trial [1].” (page 5, lines 157-159) seems out of context. The best imaging modality for PCI guidance does not appear to be the scope of the article.
You are correct but, as you see from the comments of Ref.1, this is a major issue in comparative coronary evaluations and the quoted paper indicates that OCT has a “superior spatial resolution”. Thus, we had to give some ground to the choice of the method for the invasive coronary evaluation to be compared to TTE.
Round 2
Reviewer 1 Report
In their rebuttal, the authors made few changes, but instead chose to be nonresponsive and argumentative and provide references for their arguments that, in most cases, either did not support their arguments (did they actually read these articles?) or contradicted their arguments.
The authors now state: “Overall, our work has implications for clinical intervention studies (e.g., with lipid lowering medications), where a non-invasive technique is certainly more suitable for repeated assessments. We emphasize that we selected patients with well visible LMCA at OFDI and we examined them by TTE.” If that is the main purpose of their study, then the authors have omitted a crucial and obligatory piece: the authors must demonstrate reproducibility. As has been shown in serial IVUS studies of progression/regression in the coronary arteries, but even more importantly (and more applicable to the current paper) in the serial duplex ultrasound studies of the carotid arteries, the reproducibility of a technique is paramount and is, in fact, more important than “accuracy.” TTE is entirely technique dependent. The authors have neglected this detail. Please provide reproducibility data on the TTE and OCT analyses both in terms of image acquisition and measurement/analysis. To be clear, this must include a group of patients in whom TTE was performed at two time points – it can be another cohort of patients. And it must include both inter- and intra-observer quantitative and qualitative analyses according to current standards.
Regarding the details of their rebuttal, please note the bullet points. . . .
The authors compared TTE assessment of LMCA disease using OFDI as the gold standard. The problem is that OFDI is not the gold standard for LMCA assessment. OFDI (or OCT in general) is limited by the size of the LMCA (in some patients) and by the inability to image the ostial because of the problem with blood clearance and flushing. While the authors mention limitations of TTE and the number of patients in whom TTE was suboptimal, they do not provide similar information regarding OCT. The fact that only 25 patients were enrolled in more than 2 years highlights this issue. The conclusions represent a huge over-reach. At the most, this is a feasibility study.
Thank you for the comment. The Reviewer will appreciate that OCT evaluations certainly have limitations. However, as indicated in Ref.1 (Kang et al., 2020), there is no general consensus and, in the Introduction of the ongoing OCTIVUS study, that compares OCT and IVUS, it is rightfully pointed out that OCT has a “superior spatial resolution” and that study attempts to give a clinical meaning to this important concept. In our study, we tried to obtain a meaningful comparative evaluation between the TTE and OCT, this choice being dictated by the superior resolution of the latter. The technical criticism raised, was not a major consideration in the present study, where we simply wished to provide further information on this technology, to be eventually used in clinical intervention studies requiring multiple evaluations.
Concerning the difficulties in imaging LMCA with OCT, we agree with the Reviewer’s comment. OCT was not considered suitable for the assessment of the LMCA, because of the large coronary size and poor blood washing. The recently developed OFDI provides higher acquisition speed and larger field of view compared with prior generation time-domain OCT, which may potentially overcome previous limitations. Our team recently demonstrated that OFDI can accurately evaluate the LM and detect and assess angiographically non visualized atherosclerotic plaques, providing accurate assessment of >90% of the quadrants of the LM and the ostia of its bifurcation branches (Roule V et al, 2020).
Overall, our work has implications for clinical intervention studies (e.g., with lipid lowering medications), where a non-invasive technique is certainly more suitable for repeated assessments. We emphasize that we selected patients with well visible LMCA at OFDI and we examined them by TTE.
We have now revised the over-reaching nature of our previous conclusion: In conclusion, the wider use of the TTE for the evaluation of the wall thickness of LMCA may provide a sensitive strategy to improve risk prediction, e.g., in the case of hyperlipidemic patients. This small pilot study, limited by the access of patients with available OFDI to undergo TTE evaluation, (1) has shown agreement between TTE and OFDI in the assessment of wall thickness of the LMCA and (2) tentatively suggests that the measurement of wall thickness may predict the presence of coronary calcification; more extensive studies are required to verify this notion. The use of echography to stratify risk can also provide information on the ventricular structure, thickness and wall motion abnormalities, including aortic valve stenosis, severity and progression. In our institutions vascular monitoring by C-IMT and by TTE with a special interest for hyperlipidemic patients, has generally confirmed the associated CV risk.
References
Kang, D.Y.; Ahn, J.M.; Park, H.; Lee, P.H.; Kang, S.J.; Lee, S.W.; Kim, Y.H.; Park, S.W.; Kim, S.W.; Hur, S.H., et al. Comparison of optical coherence tomography-guided versus intravascular ultrasound-guided percutaneous coronary intervention: Rationale and design of a randomized, controlled OCTIVUS trial. Am Heart J 2020, 228, 72-80, doi:10.1016/j.ahj.2020.08.003.
Roule V, Rebouh I, Lemaitre A, Bignon M, Ardouin P, Sabatier R, Labombarda F, Blanchart K, Beygui F. Evaluation of Left Main Coronary Artery Using Optical Frequency Domain Imaging and Its Pitfalls. J Interv Cardiol. 2020 Jun 12;2020:4817239. doi: 10.1155/2020/4817239. PMID: 32581660; PMCID: PMC7306070.
- Please explain how an article describing the protocol for a randomized OCT vs IVUS-guided LMCA stent implantation study -- with the primary endpoint of the non-inferiority of OCT vs IVUS -- is applicable to the current article that has nothing to do with stent implantation and, in particular, to my critique. Furthermore, the single throwaway sentence belies the fact that the authors of this article/protocol (Kang et al) used IVUS as the predicate gold-standard and the fact that there are more than 15 studies comparing IVUS-guided LMCA stenting to angiography-guided LMCA stenting, but none comparing OCT-guided LMCA stenting.
- There are 25 years of data on IVUS assessment of the LMCA including serial IVUS. There is no equivalent data. Furthermore, OCT assessment of the LMCA is associated with more artifacts than OCT assessment of non-LMCA arteries. The authors must justify using OCT as their gold-standard in the face of the imbalance in previous publications.
- When the authors state in their rebuttal “The technical criticism raised, was not a major consideration in the present study, where we simply wished to provide further information on this technology, to be eventually used in clinical intervention studies requiring multiple evaluations,” are they referring to TTE or to OCT? If TTE, then reproducibility of their methodology becomes even more important. See the beginning of my critique.
The patient population is highly skewed toward STEMI/NSTEMI patients. If the purpose of TTE assessment of the LMCA is to identify patients with CAD who would benefit from risk factor control, they should include such patients. STEMI/NSTEMI patients obviously have CAD.
As above indicated, the study followed the opposite route, i.e., that of recruiting patients with well visible OFDI and with ischemic disease (80% STEMI, 16% NSTEMI) as reported in Table 1. This is what we found. There is no question such patients need risk factor control.
- So this technique is only relevant to the subset of patients in whom both OCT and TTE are adequate? How few is this?
How did the authors image the ostium of the LMCA with OCT? The authors stated: "The whole LMCA (from the ostium or catheter tip to the ostia of its bifurcation branches defined as the first 5 mm of the left anterior descending artery and/or left circumflex artery) was analysed." I am skeptical that they were able to image the ostium of the LMCA in 32 patients. Also, the authors never provided the methodology used to measure LMCA wall thickness measurements; and they should provide a more complete output of the OCT analysis of the LMCA and proximal LAD.
The OFDI procedures were performed according the methods described in the Expert Review Document published by Prati F et al.
We have now added some details regarding the OFDI method as follows: ‘All images were recorded digitally, stored, and each frame read off-line by 2 investigators (VR and IR) using previously validated criteria for OFDI plaque characterization (Jia et al., 2013). We analyzed the whole LMCA from the LMCA ostium or catheter tip to the ostia of its bifurcation branches, defined as the first 5mm of the left anterior descending artery and/or left circumflex artery. LM length was obtained from the OFDI longitudinal reconstructions and defined as the distance between the first distal frame of the LM at cross sectional image and the last proximal LM frame before aorta or catheter visualization. When atherosclerotic plaque was identified, at least 3 measurements of the intima and media thickness were performed where the plaque was largest.’
References
Prati F, Regar E, Mintz GS, Arbustini E, Di Mario C, Jang IK, Akasaka T, Costa M, Guagliumi G, Grube E, Ozaki Y, Pinto F, Serruys PW; Expert's OCT Review Document. Expert review document on methodology, terminology, and clinical applications of optical coherence tomography: physical principles, methodology of image acquisition, and clinical application for assessment of coronary arteries and atherosclerosis. Eur Heart J. 2010 Feb;31(4):401-15. doi: 10.1093/eurheartj/ehp433. Epub 2009 Nov 4. PMID: 19892716.
Jia H, Abtahian F, Aguirre AD, Lee S, Chia S, Lowe H, Kato K, Yonetsu T, Vergallo R, Hu S, Tian J, Lee H, Park SJ, Jang YS, Raffel OC, Mizuno K, Uemura S, Itoh T, Kakuta T, Choi SY, Dauerman HL, Prasad A, Toma C, McNulty I, Zhang S, Yu B, Fuster V, Narula J, Virmani R, Jang IK. In vivo diagnosis of plaque erosion and calcified nodule in patients with acute coronary syndrome by intravascular optical coherence tomography. J Am Coll Cardiol. 2013 Nov 5;62(19):1748-58. doi: 10.1016/j.jacc.2013.05.071. Epub 2013 Jun 27. PMID: 23810884; PMCID: PMC3874870.
- The Prati article did NOT address the use of OCT (or OFDI or FD-OCT) in the assessment of LMCA disease and in particular, the “methods described in the Expert Review Document published by Prati F et al.” It certainly did not address the difficulties in assessing the aorto-ostial junction with OCT (or OFDI or FD-OCT). To the contrary, the Prati article highlighted these limitations and proposed no solution.
- The authors state: “LM length was obtained from the OFDI longitudinal reconstructions and defined as the distance between the first distal frame of the LM at cross sectional image and the last proximal LM frame before aorta or catheter visualization.” In other words, the ostium was not always visualized. The authors neglected to provide the information as to how often the aorto-ostium was visualized or not visualized and a comparison of the TTE vs OCT assessment of the LMCA ostium. As important is their statement about catheter visualization. If the catheter is in the LMCA, that portion of the LMCA is obscured. The authors neglected to provide the information as to how much of the proximal LMCA was obscured because this can be a substantial part of the proximal LMCA making measurement of LMCA length and complete assessment of LMCA plaque composition unreliable.
How complete was the TTE assessment of the LMCA? Did it always include the bifurcation? It is noteworthy that there is a lot of detail regarding TTE, but a paucity of data regarding OCT methodology. There are no angiographic methods.
TTE assessment: The bifurcation was always included. We revised the sentence as follows: ‘After the origin of the LMCA was identified, the image was optimized by step-by-step movement of the transducer according to the course of the vessel in order to visualize the LMCA in its entire extension including the bifurcation’ (Figure 1B).
Angiographic methods: we revised the sentence in the Methods section as follows: “Quantitative Coronary Angiography. Quantitative coronary angiography was performed offline (CAAS II,Pie Medical, Maastricht, Netherlands) using validated quantitative methods (Garrone et al., 2009).
Reference
Garrone P, Biondi-Zoccai G, Salvetti I, Sina N, Sheiban I, Stella PR, Agostoni P. Quantitative coronary angiography in the current era: principles and applications. J Interv Cardiol. 2009 Dec;22(6):527-36. doi: 10.1111/j.1540-8183.2009.00491.x. Epub 2009 Jul 13. PMID: 19627430.
- Citing an article is not the same as providing methodology. The Garrone article does not address the use of QCA to assess LMCA disease which has its unique problems and limitations. More importantly, the revised article does not present any definitions, criteria, or methodology for the angiographic characteristics reported in Table 1. What were the criteria of single-vessel, 2-vessel, or 3-vessel disease? What were the criteria for “Angiographic signs of LMCA atherosclerosis: No Lesions or Lesions <30%”? A diffusely diseased LMCA can appear to be disease-free angiographically.
The authors second purpose is "to associate wall thickness of LMCA with coronary calcium as determined by OFDI." However, they never report OCT coronary calcium. They do report the association between TTE wall thickness and fibrocalcific plaque, but not coronary calcium. In addition, they never report OCT quantification of either coronary calcium or fibrocalcific plaque. Finally, there were too few patients without fibrocalcific plaque (n=6) for this type of statistical comparison. The authors stated: "The diagnostic 1performance of wall thickness of the LMCA for the detection of fibro-calcific plaque is presented as a ROC curve (Figures 1C). A value >1.4 mm was associated with fibro-calcific plaque (area under the curve: 0.815; p=0.006), with the highest sensitivity and specificity of 76.9% and 80%, respectively, to predict fibro-calcific plaques of the LMCA. Youden’s Index was 0.56." Where was the fibrocalcific plaque? In the LMCA or elsewhere? Why is this important? If there is fibrocalcific plaque in the LMCA or increased LMCA all thickness, doesn't the patient have CAD?
Studied patients had definite CAD Since one of us (GW) had reported that in hypercholesterolemic pts with CVD there was calcium deposition and this might be linked to raised CV risk (Lahti et al., 2019), we tested the hypothesis, that turned out to be probably right, that coronary calcification is associated with raised LMCA thickness by TTE. The predictive role of raised calcium deposition is supported by a larger study evaluating long term CV risk (Garcia-Garcia et al., 2014) having to do specifically with the deposition of calcium in the LMCA, not elsewhere.
The limits of this type of statistical evaluation are well known. Actually, in the paper in Ref. 2 (Ghanem et al., 2019), where coronary thickness was determined by MRI, a threshold of 1.41 mm is reported for the presence of CAD. In this paper the Authors repeatedly underline the correlation of CAD scores with calcium deposition.
In addition, interest is focusing on non-stenotic, spotty calcifications and attenuation that correlate with markers of plaque instability. Most recently the issue of coronary calcification has grown in interest after the observation that a calcium score of 0 in familial hypercholesterolemic patients was associated with no rise in CV risk. (Mszar 2020 (a) and Mszar 2020 (b)). The text has been revised according and now reads as follows “The absence of coronary calcium can provide a potentially useful risk marker: a CAC= 0 in a patient with familial hypercholesterolemia appears to be associated with essentially no rise in coronary risk [22,23], providing a clinically more significant information compared to polygenic risk scores [24].”
Reference
Lahti, S.J.; Feldman, D.I.; Dardari, Z.; Mirbolouk, M.; Orimoloye, O.A.; Osei, A.D.; Graham, G.; Rumberger, J.; Shaw, L.; Budoff, M.J., et al. The association between left main coronary artery calcium and cardiovascular-specific and total mortality: The Coronary Artery Calcium Consortium. Atherosclerosis 2019, 286, 172-178, doi:10.1016/j.atherosclerosis.2019.03.015.
Garcia-Garcia, H.M.; Muramatsu, T.; Nakatani, S.; Lee, I.S.; Holm, N.R.; Thuesen, L.; van Geuns, R.J.; van der Ent, M.; Borovicanin, V.; Paunovic, D., et al. Serial optical frequency domain imaging in STEMI patients: the follow-up report of TROFI study. Eur Heart J Cardiovasc Imaging 2014, 15, 987-995, doi:10.1093/ehjci/jeu042.
Ghanem, A.M.; Matta, J.R.; Elgarf, R.; Hamimi, A.; Muniyappa, R.; Ishaq, H.; Hadigan, C.; McConnell, M.V.; Gharib, A.M.; Abd-Elmoniem, K.Z. Sexual Dimorphism of Coronary Artery Disease in a Low- and Intermediate-Risk Asymptomatic Population: Association with Coronary Vessel Wall Thickness at MRI in Women. Radiol Cardiothorac Imaging 2019, 1, e180007, doi:10.1148/ryct.2019180007.
[22] Mszar, R.; Nasir, K.; Santos, R.D. Coronary Artery Calcification in Familial Hypercholesterolemia: An Opportunity for Risk Assessment and Shared Decision Making With the Power of Zero? Circulation 2020, 142, 1405-1407, doi:10.1161/CIRCULATIONAHA.120.049057.
[23] Mszar, R.; Grandhi, G.R.; Valero-Elizondo, J.; Virani, S.S.; Blankstein, R.; Blaha, M.; Mata, P.; Miname, M.H.; Al Rasadi, K.; Krumholz, H.M., et al. Absence of Coronary Artery Calcification in Middle-Aged Familial Hypercholesterolemia Patients Without Atherosclerotic Cardiovascular Disease. JACC Cardiovasc Imaging 2020, 13, 1090-1092, doi:10.1016/j.jcmg.2019.11.001.
[24] Wunnemann, F.; Sin Lo, K.; Langford-Avelar, A.; Busseuil, D.; Dube, M.P.; Tardif, J.C.; Lettre, G. Validation of Genome-Wide Polygenic Risk Scores for Coronary Artery Disease in French Canadians. Circ Genom Precis Med 2019, 12, e002481, doi:10.1161/CIRCGEN.119.002481.
- The authors neglected to address the vast majority of my points and provide any of the additional information requested.
Author Response
The authors now state: “Overall, our work has implications for clinical intervention studies (e.g., with lipid lowering medications), where a non-invasive technique is certainly more suitable for repeated assessments. We emphasize that we selected patients with well visible LMCA at OFDI and we examined them by TTE.”
We apologize for a misunderstanding. This statement was only in the answer to the reviewer’s prior Comments, and not included in the revised text. In the revised text we simply mentioned that the method may “provide a sensitive strategy to improve risk prediction”. We consider on the basis of our pilot study that this is not an unreasonable assertion, but should be willing to delete it if the reviewer insists. We have also now removed the wording on “hyperlipidemic patients” and the text now reads as follows “In conclusion, the wider use of the TTE for the evaluation of the wall thickness of LMCA may provide a sensitive approach to improve risk prediction in patients who require further evaluation.”
If that is the main purpose of their study, then the authors have omitted a crucial and obligatory piece: the authors must demonstrate reproducibility. As has been shown in serial IVUS studies of progression/regression in the coronary arteries, but even more importantly (and more applicable to the current paper). In the serial duplex ultrasound studies of the carotid arteries, the reproducibility of a technique is paramount and is, in fact, more important than “accuracy.” TTE is entirely technique dependent. The authors have neglected this detail. Please provide reproducibility data on the TTE and OCT analyses both in terms of image acquisition and measurement/analysis. To be clear, this must include a group of patients in whom TTE was performed at two time points – it can be another cohort of patients. And it must include both inter- and intra-observer quantitative and qualitative analyses according to current standards.
We do not have any chance to provide data on patients in whom TTE was performed at two time points. You have to consider this as a simple pilot study (this has been stated). Moreover, since we are aware of this limitation, it has been now listed among limitations. The text now reads as follows
”Finally, reproducibility of wall thickness of the LMCA by TTE in another group of coronary individuals should be tested.”
We can provide the operator’s reproducibility in the below reported Table. This was also requested by Reviewer 2.
Variables |
Intra observer reproducibility Intra Class correlation (95% Confident Interval)
|
Inter observer reproducibility Intra Class Correlation (95% Confident Interval) |
Anterior thickness |
0.98 (0.88 - 0.98) |
0.93 (0.76 - 0.99) |
Internal diameter |
0.96 (0.85 - 0.99) |
0.91 (0.80- 0.96) |
A similar reproducibility for coronary TTE evaluation was reported by Perry et al. in their study on the left anterior descending coronary (Perry et al, Am J Cardiol. 2008 Apr 1;101(7):937-40).
Please explain how an article describing the protocol for a randomized OCT vs IVUS-guided LMCA stent implantation study -- with the primary endpoint of the non-inferiority of OCT vs IVUS -- is applicable to the current article that has nothing to do with stent implantation and, in particular, to my critique. Furthermore, the single throwaway sentence belies the fact that the authors of this article/protocol (Kang et al) used IVUS as the predicate gold-standard and the fact that there are more than 15 studies comparing IVUS-guided LMCA stenting to angiography-guided LMCA stenting, but none comparing OCT-guided LMCA stenting.
The thanks the reviewer for elucidating the context of the paper by Kang et al. We agree that we quoted it out of context and have now replaced it with by Kubo et al (Eur Heart J. 2017 Nov 7;38(42):3139-3147). This paper gives details on the OFDI guided PCI compared with IVUS-guided PCI. It concludes that both OFDI and IVUS guided procedures yielded excellent angiographic and clinical results. We believe this paper supports the approach selected for the present study.
There are 25 years of data on IVUS assessment of the LMCA including serial IVUS. There is no equivalent data. Furthermore, OCT assessment of the LMCA is associated with more artifacts than OCT assessment of non-LMCA arteries. The authors must justify using OCT as their gold-standard in the face of the imbalance in previous publications. When the authors state in their rebuttal “The technical criticism raised, was not a major consideration in the present study, where we simply wished to provide further information on this technology, to be eventually used in clinical intervention studies requiring multiple evaluations,” are they referring to TTE or to OCT? If TTE, then reproducibility of their methodology becomes even more important. See the beginning of my critique. ints – it can be another cohort of patients. And it must include both inter- and intra-observer quantitative and qualitative analyses according to current standards.
The use of OCT versus IVUS is becoming more popular in cardiological practice, because of the former has greater potential to provide more detailed morphological and qualitative information on coronary plaques (Kubo et al. Eur Heart J. 2017 Nov 7;38(42):3139-3147).
However, the Reviewer is correct. OCT was not considered suitable for the assessment of the LM because of the large coronary size and poor blood washing, which are responsible of artefact especially when assessing the proximal segments of the LMCA. Nevertheless, we demonstrated recently that OFDI is more accurate for the evaluation of LMCA (Roule V et al, J Interv Cardiol. 2020 Jun 12;2020:4817239. doi: 10.1155/2020/4817239). Our study showed that OFDI can accurately evaluate the LMCA and detect and assess angiographically unvisualized atherosclerotic plaques, providing accurate assessment of >90% of the quadrants of the LMCA and the ostia of its bifurcation branches. Most artifacts were located in the proximal LM and their rate decreased distally (the area where we made our echocardiographic measurement).
So this technique is only relevant to the subset of patients in whom both OCT and TTE are adequate? How few is this?
In the response to the previous Comments by Ref. 2 we stated that “These are patients who had a positive angiography with OFDI and then had a TTE. This cuts down numbers considerably”. We concede that the sample size is accordingly small and referred to this limitation of our pilot study in the Limitations of the Discussion. This does not imply that the TTE technology will only be used in practice in coronary patients after OFDI, and can expand to clarify this further if the reviewer wishes.
The Prati article did NOT address the use of OCT (or OFDI or FD-OCT) in the assessment of LMCA disease and in particular, the “methods described in the Expert Review Document published by Prati F et al.” It certainly did not address the difficulties in assessing the aorto-ostial junction with OCT (or OFDI or FD-OCT). To the contrary, the Prati article highlighted these limitations and proposed no solution.
We thank the reviewer for pointing this error. We apologize for this oversight in our response to you. The manuscript by Prati was not quoted in the manuscript.
The authors state: “LM length was obtained from the OFDI longitudinal reconstructions and defined as the distance between the first distal frame of the LM at cross sectional image and the last proximal LM frame before aorta or catheter visualization.” In other words, the ostium was not always visualized. The authors neglected to provide the information as to how often the aorto-ostium was visualized or not visualized and a comparison of the TTE vs OCT assessment of the LMCA ostium. As important is their statement about catheter visualization. If the catheter is in the LMCA, that portion of the LMCA is obscured. The authors neglected to provide the information as to how much of the proximal LMCA was obscured because this can be a substantial part of the proximal LMCA making measurement of LMCA length and complete assessment of LMCA plaque composition unreliable.
We thank the Reviewer for this comment. As above reported, we agree that OCT was not considered suitable for the assessment of the LMCA because of the large coronary size and poor blood washing. Nevertheless, dedicated studies testing the possible pitfalls of OFDI imaging in LMCA are scarce, especially in a population without previous LM stenting. However, we recently conducted a study to assess the quality of ODFI in order to analyze its potential artefacts in the assessment of LMCA arterial wall in coronary artery disease patients with or without detectable angiographic LMCA lesions (Roule V et al, J Interv Cardiol. 2020 Jun 12;2020:4817239. doi: 10.1155/2020/4817239). This study showed that OFDI can accurately evaluate the LMCA and detect and assess angiographically unvisualized atherosclerotic plaques, providing accurate assessment of >90% of the quadrants of the LMCA and the ostia of its bifurcation branches. Most artifacts were located in the proximal LM and their rate decreased distally (the area where we made our echocardiographic measurement). Even though, out-of-field and residual blood-related artifacts should be considered when using OFDI in the ostial or proximal LMCA, we estimate that OFDI can accurately evaluate LMCA bifurcation, which is the most commonly diseased segment for LM stenosis, and provides a precise evaluation of LMCA atherosclerotic plaques.
Citing an article is not the same as providing methodology. The Garrone article does not address the use of QCA to assess LMCA disease which has its unique problems and limitations. More importantly, the revised article does not present any definitions, criteria, or methodology for the angiographic characteristics reported in Table 1. What were the criteria of single-vessel, 2-vessel, or 3-vessel disease?
Thank you for pointing this out. We have accordingly removed this reference and updated “2.2 Optical frequency domain imaging studies” section as follows “The OFDI procedure (Lunawave®, FastView®, Terumo Europe, Belgium), performed as previously described [11], carried out after intracoronary administration of nitroglycerin (0.2 mg). The images were acquired out by using a non-occlusive technique (rate of 160 frames/s) during an automated pullback of the catheter (speed of 20 mm/s). The pullback was performed during continuous intracoronary injection of contrast medium through the guiding catheter using an injection pump (flow rate of 4 mL/s, maximum of 3 s). All images were recorded digitally, stored, and each frame read offline by 2 investigators (VR and IR) using previously validated criteria for OCT plaque characterization [12]. We analysed the whole LMCA from the ostium or catheter tip to the ostia of its bifurcation branches defined as the first 5mm of the left anterior descending artery and/or left circumflex artery. LM length was obtained from OFDI longitudinal reconstructions and defined as the distance between the first distal frame of the LM at cross sectional image and the last proximal LM frame before aorta or catheter visualisation. When an atherosclerotic plaque was identified, at least 3 measurements of the intima and media thickness were performed where the plaque was largest. Quantitative coronary angiography was performed offline (CAAS II,Pie Medical, Maastricht, Netherlands) using validated quantitative methods.”
Relative to single-vessel or multivessel disease, a few lines have been added in the results section. These read as follows “Most of the patients presented with ST-elevation myocardial infarction (n= 20, 80 %) and were characterized by a single vessel disease (n=17, 68%); 32% had angiographically multivessel diseases.”
What were the criteria for “Angiographic signs of LMCA atherosclerosis: No Lesions or Lesions <30%”? A diffusely diseased LMCA can appear to be disease-free angiographically.
We apologies for not making this clear. The criterion used to define no angiographic atherosclerosis is the absence of detected atherosclerotic lesions and hence “no lesion”. In Table 1, the notation of < 30% is obviously wrong. It has been corrected with > 30%. We have now made this clearer in the text. The results now read as follows “Angiographic signs of atherosclerosis of LMCA, i.e., lesions ³ 30%, were detected in 40% of patients.”
The authors neglected to address the vast majority of my points and provide any of the additional information requested.
We respectfully acknowledge the expertise of the reviewer and have attempted to meet the requests for revisions of the paper. We hope our new changes meet with approval and would be willing to make further changes as requested.

Reviewer 2 Report
Please on line 188 explain the acronym ASCDV. I have no other comments to do.Author Response
We thank the reviewer for pointing this out. The acronym, which appears ones, has been deleted and replaced with atherosclerotic cardiovascular disease.
Reviewer 3 Report
I have no further comments.
Author Response
We are most than grateful